# Knowledge-Selective Pretraining for Attribute Value Extraction

**Hui Liu[1], Qingyu Yin[2]\*, Zhengyang Wang[2], Chenwei Zhang[2],**
**Haoming Jiang[2], Yifan Gao[2], Zheng Li[2], Xian Li[2], Chao Zhang[2],**
**Bing Yin[2], William Yang Wang[2], Xiaodan Zhu[1]**
[1]Ingenuity Labs Research Institute & ECE, Queen's University, Canada
[2]Amazon.com Inc, Palo Alto, CA, USA
{hui.liu,xiaodan.zhu}@queensu.ca
{qingyy,zhengywa,cwzhang,jhaoming,yifangao}@amazon.com

## Abstract

Attribute Value Extraction (AVE) aims to retrieve the values of attributes from the product profiles. The state-of-the-art methods tackle the AVE task through a question-answering (QA) paradigm, where the value is predicted from the context (i.e. product profile) given a query (i.e. attributes). Despite of the substantial advancements that have been made, the performance of existing methods on rare attributes is still far from satisfaction, and they cannot be easily extended to unseen attributes due to the poor generalization ability. In this work, we propose to leverage pretraining and transfer learning to address the aforementioned weaknesses. We design a **K**nowledge-**Sel**ective **F**ramework (KSelF) based on query expansion that can be closely combined with a pretraining corpus to boost the performance. Meanwhile, considering the public AE-pub dataset contains considerable noise, we construct a larger benchmark EC-AVE collected from E-commerce websites. We conduct evaluation on both of these datasets. The experimental results demonstrate that our proposed KSelF achieves new state-of-the-art performance without pretraining. When incorporated with the pretraining corpus, the performance of KSelF can be further improved, particularly on the attributes with limited training resources.

## 1 Introduction

With the fast expansion of E-commerce systems, Attribute Value Extraction (AVE), which aims to retrieve the values of attributes from the product profiles, has attracted significantly more research attention since it can be used to supplement product information and hence serve better product search (Xiao et al., 2021; Luo et al., 2022), recommendations (Hwangbo et al., 2018), and other E-commerce tasks. As shown in Figure 1, the goal of AVE is to extract values of different attributes, e.g.,

---
*corresponding author

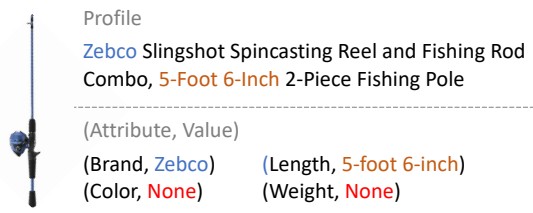

> Profile
> Zebco Slingshot Spincasting Reel and Fishing Rod Combo, 5-Foot 6-Inch 2-Piece Fishing Pole
>
> (Attribute, Value)
> (Brand, Zebco)      (Length, 5-foot 6-inch)
> (Color, None)       (Weight, None)

Figure 1: Example of (*attribute*, *value*) pairs of a product. For the attributes that are not presented in the profile, the values are "None".

"Zebco" from the profile for the attribute *Brand*. Automating this process can considerably enrich the product information with fewer human efforts, which has become a hot research topic for many E-commerce stores (Xu et al., 2019).

Existing research on AVE mainly falls into two paradigms: (1) sequence-tagging, and (2) question answering (QA). The sequence-tagging paradigm formulates AVE as a named entity recognition problem, where the model is built to identify the attribute types of the tokens in product profiles (Putthividhya and Hu, 2011; More, 2016; Zheng et al., 2018; Xu et al., 2019; Liang et al., 2020; Zhu et al., 2020; Jiang et al., 2021; Wang et al., 2022). However, the sequence-tagging methods suffer from the limitations that they do not scale to cope with a large number of attributes and cannot be extended to unseen attributes (Wang et al., 2020). Hence, recent works have proposed to formulate AVE as a QA task, where the product profile and attribute are taken as the context and query, respectively (Wang et al., 2020; Shinzato et al., 2022; Ding et al., 2022; Yang et al., 2022), and the model is trained to extract the attribute value span from the context. The QA paradigm has shown to be more flexible and scalable than sequence-tagging methods, achieving state-of-the-art performance on the AVE task.

While the QA paradigm has yielded improved results, they still face significant limitations that need to be addressed. Previous research demonstrates that the QA-based methods do not perform well on

the attributes with limited training resources (Wang et al., 2020; Shinzato et al., 2022), even though Shinzato et al. (2022) propose to build better representations by expanding the query with values of the same attribute from the training set and accordingly achieves promising overall performance. The sub-optimal performance on attributes with limited training resources hinders the practical application of QA-based AVE methods to E-commerce, because new and unseen attributes continue to arise with the introduction of new products.

To overcome the aforementioned problems, we develop a task-specific pretraining method for the QA-based AVE, which provides the model strong generalization capabilities, allowing the model to quickly adapt to rare and unseen attributes. Inspired by the recent popularity of pretrained language models (Devlin et al., 2019; Liu et al., 2019; Chen et al., 2020; He et al., 2021) that leverage large-scale unlabeled corpora from the Web to perform pretraining, we collect product profiles as well as the corresponding product attributes from various public E-commerce websites. To automatically and efficiently construct data for pretraining, we conduct string match in the collected product information, and obtain the large-scale corpus consisting of (*profile*, *attribute*, *value*) triples. Using these data, we further pretrain a public language model for an AVE task employing the QA paradigm.

To further improve the performance, we adopt the query expansion technique. Given a product and an attribute to be filled with a value, Shinzato et al. (2022) employs values of the same attributes retrieved from training corpus as knowledge to expand the query, improving upon previous methods that only regard the attributes as queries. The assumption behind their method is that such expanded knowledge provides informative hints on helping identify the correct value span of the target attribute. However, these retrieved values constitute a vast set which may include those values that are not closely related to the current product, thus providing little hints and even introducing unexpected noise. For example, the value of attribute "material" for a "shirt" has nothing to do with that for a "desk". This is particularly severe in the pretraining scenario because the pretraining data is typically in large-scale and there will be a huge number of helpless values from a variety of products for the same attribute. Ideally, a desirable solution should be able to effectively identify the

usefulness of knowledge and employ informative knowledge for query expansion. Therefore, on the basis of query expansion-based AVE, we propose a **K**nowledge-**Sel**ective **F**ramework (KSelF). When performing query expansion, the proposed method first categorizes the retrieved values into the *more-related* and *less-related*, based on their similarity to the current product. Then we construct the expanded queries by properly exploiting the two types of attributes during training and inference. KSelF can be closely combined with the constructed pretraining corpus, allowing for more efficient utilization of training data, which further improves the generalization ability of the model.

To benchmark the KSelF, we conduct experiments on the AE-pub dataset (Xu et al., 2019). However, we notice that AE-pub (Xu et al., 2019) contains significant noise. To encourage future research on this task, we further propose a new benchmark EC-AVE of high quality collected from a well-structured E-commerce website. The results demonstrate that our proposed KSelF achieves new state-of-the-art performance on both datasets without pretraining. When pretrained on the large-scale corpus and then finetuned on the downstream datasets, KSelF can further attain better performance, particularly on the attributes with limited training resources. We summarize our contributions as follows:

- We propose a novel pretraining method for QA-based AVE model that improves model performance on rare and unseen attributes.

- We propose a knowledge-selective query expansion framework that is capable of more efficiently exploiting the training data.

- We conduct extensive experiments on the two AVE datasets, demonstrating that our method achieves state-of-the-art performance.

## 2 Related Work

### 2.1 Attribute Value Extraction

The goal of Attribute Value Extractive (AVE) is to extract the values of attributes from the product profile. Early works adopt rule-based techniques including domain-specific dictionary or vocabulary to identify the important phrases and attributes (Gopalakrishnan et al., 2012; Vandic et al., 2012). With the wide application of deep learning, recent works mainly adopt two paradigms for the AVE

task: (1) sequence tagging and (2) question answering. For the sequence tagging paradigm, the main idea to is train a model to predict the attribute types of tokens in the product profile sequentially (Putthividhya and Hu, 2011; Shinzato and Sekine, 2013; More, 2016). To achieve this goal, different sequence-tagging-based approaches have been proposed. Zheng et al. (2018) build an end-to-end BiLSTM-CRF model with separate tag sets for different attributes, while Xu et al. (2019) adopt BERT-CRF and propose to adopt a global set of tags for all the attributes, which enables the model to fit larger attribute sets. Kumar and Saladi (2022) further use reinforcement learning to enhance the performance of sequence-tagging models.

However, the sequence-tagging paradigm can only be used for a fixed attribute set, and thus suffers from the lack of scalability and generalizability. To extend the model to large-scale attribute sets, Wang et al. (2020) utilize the question-answering paradigm where the product profile and the attribute are respectively regarded as context and question. The model AVEQA is proposed which concatenates the profile and the attribute into a single query string, and then predicts the value span from the profile. To encourage the models to fully utilize the advantage of the QA paradigm, Shinzato et al. (2022) expand the query string with values of the same attribute from the training set, since these values can serve as the knowledge to learn better query representations. Despite of the promising performance achieved by the QA-based models, they are shown to perform poorly on the attributes that are not seen in the training set, demonstrating sub-optimal generalization on the new attributes. We hence build our method upon the QA paradigm and propose a knowledge-selective query expansion framework to encourage the model to more efficiently exploit the training data.

## 2.2 Pretrained Language Models

Pretrained language models have recently embraced great success in a variety of NLP tasks (Devlin et al., 2019; Radford et al., 2019; Dai et al., 2019; Yang et al., 2019; Liu et al., 2019; Chen et al., 2020; He et al., 2021). They are pretrained with large-scale corpus and then adapted to different downstream tasks or datasets from various domains, showing good generalization ability. However, the existing pretrained models lack the ability to predict a value span from the context conditioned on

a query, hence they cannot be directly applied to the AVE task, In this work, we build the pretraining corpus collected from the Web and pretrain the model with the constructed corpus, which, to the best of our knowledge, is the first work that applies the idea of pretraining to AVE.

## 3 Background

### 3.1 Task Definition

Given a product profile $\boldsymbol{x} = \{x_1, x_2, \cdots, x_n\}$ and an attribute $\boldsymbol{a} = \{a_1, a_2, \cdots, a_m\}$ where $n$ and $m$ are the number of tokens in the profile and the attribute respectively, the goal of AVE is to predict the value of the current attribute, which is a single text span in $\boldsymbol{x}$ with beginning position as $P_b$ and ending position as $P_e$ ($1 \leq P_b \leq P_e \leq n$). If $\boldsymbol{x}$ does not contain a value span for the current attribute, which we call as a *negative* instance in this paper, a special value *None* should be predicted.

### 3.2 Query Expansion for QA-Based AVE

Consider the product profile $\boldsymbol{x}$ and the attribute $\boldsymbol{a}$. A vanilla query is constructed as:

$$\boldsymbol{q} = [\text{CLS}; \boldsymbol{x}; \text{SEP}; \boldsymbol{a}] \quad (1)$$

$\boldsymbol{q}$ will be fed into a pretrained BERT (Devlin et al., 2019), and the contextualized vector representation of [CLS] and $\boldsymbol{x}$ from the last layer will be attained respectively as $h_{\text{CLS}} \in R^d$ and $H_{\boldsymbol{x}} \in R^{n \times d}$, where $d$ is the hidden size. Then the two vector representations will be concatenated as $H = [h_{\text{CLS}}; H_{\boldsymbol{x}}]$, and further be used to predict the starting position and the ending position as:

$$P_b = \arg\max_i(\text{Softmax}(W_b H^i))$$
$$P_e = \arg\max_{i \geq P_b}(\text{Softmax}(W_e[H^i; H^{P_b}]))$$

where $H^i$ is the contextual embedding of the $i^{th}$ token in $H$. $W_b$ and $W_e$ are parameters that map the embedding to output logits. For the negative instances where there is no value presented in the product profile, the ground truth of $P_b$ and $P_e$ is set to be 0, which is the position of the [CLS] token.

To provide better representation of the attribute, Shinzato et al. (2022) propose several techniques to expand Eq. 1. Admittedly, values of an attribute can be used for illustrating the attribute. Shinzato et al. (2022) propose to utilize the values of $\boldsymbol{a}$ that appear in the training data as the run-time knowledge as $\boldsymbol{v_a} = [v_{\boldsymbol{a},1}; \text{SEP}; v_{\boldsymbol{a},2}; \text{SEP}; \cdots]$

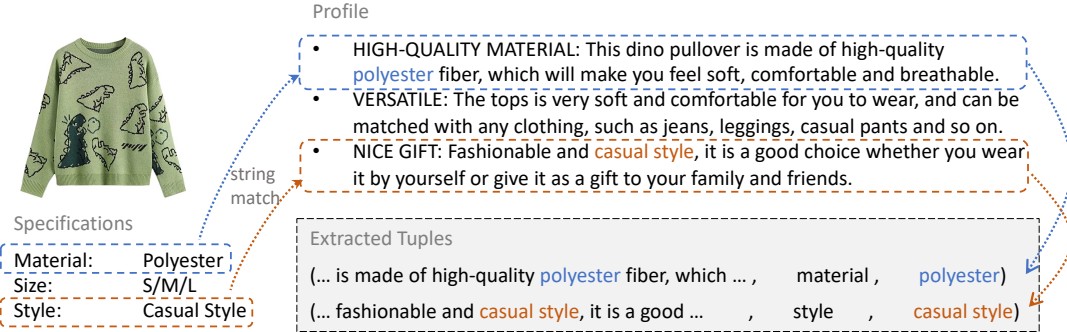

Figure 2: Process of harvesting (*profile*, *attribute*, *value*) triples from product profile and their specifications.

where $v_{a,i}$ is a seen value of attribute $a$ that appears in the training data. Hence, Eq. 1 can be expanded with the run-time knowledge as $q_e = [\text{CLS}; x; \text{SEP}; a; \text{SEP}; v_a]$. Another technique they adopt is domain token mixing (Britz et al., 2017) by using two special tokens SEEN and UNSEEN to denote if run-time knowledge is provided in the query. Hence, for an example with profile $x$ and attribute $a$, two queries can be built for model training:

$$q_s = [\text{CLS}; x; \text{SEP}; \text{SEEN}; a; \text{SEP}; v_a]$$
$$q_u = [\text{CLS}; x; \text{SEP}; \text{UNSEEN}; a]$$

During the inference stage, SEEN and UNSEEN tokens are adopted for seen attributes and unseen attributes, respectively.

## 4 Method

### 4.1 Pretraining Data Construction

Here, we introduce how to build the pretraining corpus, of which the overall construction process is shown in Figure 2. In general, we collect the pretraining corpus from various E-commerce websites. The construction process contains two-stages: (1) product profile collection and (2) triples extraction.

**Product Profile Collection** When presenting a product, an E-commerce website often includes both the product descriptions and the corresponding product specifications. The product descriptions are some sentences that describe the details or features of a product, while the specifications consist of some product attributes and the corresponding values, as shown in the examples in Figure 4 of Appendix D. We visit multiple E-commerce websites and for each product, we crawl its descriptions and specifications, if any. As such, we obtain 22M products with their corresponding descriptions and specifications from various product categories including clothing, books, among others.

| #Triple | Length | #Attributes | #Pos. | #Neg. |
|---------|--------|-------------|-------|-------|
| 4M | 47.75 | 5270 | 2.75M | 1.25M |

Table 1: Statistics of pretraining corpus. Pos. and Neg. denote positive instances and negative instances respectively. Length denotes the average length of the profiles in the triples.

**Triple Extraction** This step aims to retrieve (*profile, attribute, value*) triples from the product descriptions and specifications collected in the last step. We denote $D = \{d_i | i \in [1, 2, \cdots, |D|]\}$ as the descriptions of a product that contain $|D|$ sentences, and $S = \{a_j : v_j | j \in [1, 2, \cdots, |S|]\}$ as the specifications that contain $|S|$ attribute-value pairs. We conduct string match between the descriptions and the specifications. If a value $v_j$ appears in a description sentence $d_i$, a triple $(d_i, a_j, v_j)$ will be retrieved where $d_i, a_j, v_j$ are the profile, attribute, and value, respectively, and will be used for pretraining our AVE model.

The triple set collected through the aforementioned two steps contains a total of 5,270 attributes, while the attributes exhibit an apparent long-tail distribution which is common in many datasets (Zhang et al., 2021). In order to mitigate the negative effects brought by the extreme long-tail distribution, we keep the maximum instance number of one attribute to 10K and perform down-sampling accordingly. Meanwhile, the triples collected with the two steps are all positive instances, while an AVE dataset may also include negative instances as discussed in Section 3.1. Hence, we randomly combine some product descriptions and attributes from the product profiles, and use the special value *None* to form a negative instance. In this way, we finally obtain a pretraining corpus of 4M triples with 5270 attributes. Table 1 presents a detailed statistics of the collected pretrain corpus.

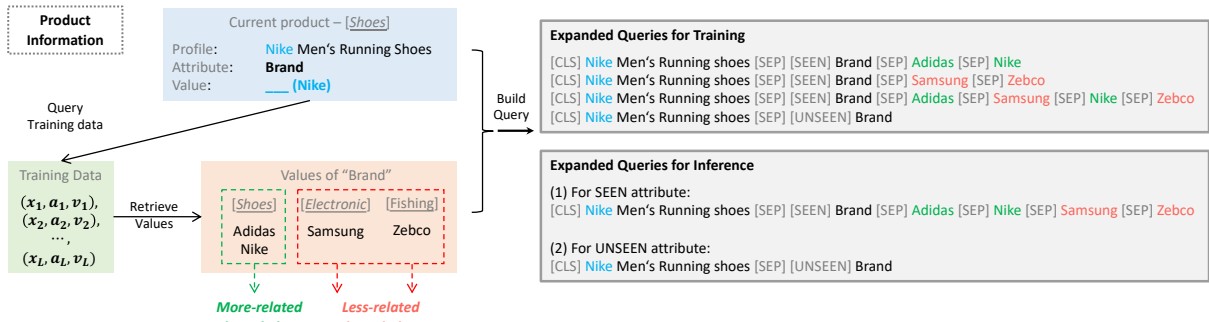

Figure 3: Overall illustration of our proposed knowledge-selective query expansion framework. Given the current product information, the goal is to predict the value of "Brand". As a first step, the values of "Brand" are retrieved from the training data as run-time knowledge, which is further separated into two subsets, more-related knowledge and less-related knowledge. Then the two subsets, together with the product information, will form different queries which are used during the training stage and the inference stage.

## 4.2 Knowledge-Selective Framework

Despite of the performance achieved by Shinzato et al. (2022) briefed in Section 3.2, there are shortcomings in their proposed method.

- First, the proposed run-time knowledge contains all the seen values of an attribute, while some of the values may be less related to the current profile $x$, and hence provide limited information in explaining the current attribute $a$. As shown in the example in Figure 3, given the current product profile "Nike Men's Running Shoes", we can retrieve various values of attribute "Brand" from the training data, including values of the *Shoes* category like "Adidas" and "Nike", and values from other categories including *Electronic* and *Fishing*. However, only the values from the *Shoes* category are highly related to the current profile and can help to explain the current attribute, while the values of *Electronic* and *Fishing* may contribute little or even become noise in the run-time knowledge.

- Second, pretrained models limit the length of the input query. The values in the run-time knowledge are ranked in descending order of frequency, which may result in the truncation of the more-related values if the query is long.

The above problems will become even more severe when the method is combined with the pretraining corpus since one attribute may contain large number of values from different product categories.

To overcome the aforementioned weaknesses, we propose a **K**nowledge-**Sel**ective **F**ramework (KSelF) based on query expansion. KSelF first measures the similarity score between profiles,

which is used to categorize the knowledge (i.e. attribute values) into more-related and less-related ones. KSelF then constructs the expanded queries by properly exploiting these two types of attributes during training and inference. Figure 3 shows the overview of the proposed framework.

**Knowledge Categorization**   Given a product profile $x$ and an attribute $a$, we first retrieve all the triples from training data of which the attribute is $a$ as $\mathcal{T} = \{(x_i, a, v_i) | i \in [1, 2, \cdots, |\mathcal{T}|]\}$. For a triple $(x_i, a, v_i) \in \mathcal{T}$, we calculate the similarity between two profiles $x$ and $x_i$, where the calculation method is differently designed for the pretraining stage and fine-tuning stage.

- For the fine-tuning stage, we conduct entity retrieval (De Cao et al., 2021) on both $x$ and $x_i$. The similarity is defined as the number of common entities of the two profiles.

- For the pretraining stage, the similarity is defined as the bag-of-word similarity of the two profiles for the consideration of both precision and computation efficiency.

More details of similarity measurements can be found in Appendix A. After obtaining the similarities between $x$ and each profile in $\mathcal{T}$, we can separate $\mathcal{T}$ into two sub-sets $\mathcal{T}_m$ and $\mathcal{T}_l$ according to a predefined threshold on the profile similarity socres. The values in $\mathcal{T}_m$ are from the triples where the profiles are more similar to $x$, which we call *more-related* knowledge, while the values in $\mathcal{T}_l$ are *less-related* knowledge.

**Query Expansion with Selected Knowledge**
We expand queries with the run-time knowledge

according to the knowledge categorization. Specifically, we have two types of expansions:

$$\boldsymbol{q}_s^m = [\text{CLS}; \boldsymbol{x}; \text{SEP}; \text{SEEN}; \boldsymbol{a}; \text{SEP}; \boldsymbol{v}_a^m]$$
$$\boldsymbol{q}_s^l = [\text{CLS}; \boldsymbol{x}; \text{SEP}; \text{SEEN}; \boldsymbol{a}; \text{SEP}; \boldsymbol{v}_a^l]$$

where $\boldsymbol{v}_a^m$ and $\boldsymbol{v}_a^l$ are the values from $\mathcal{T}_m$ and $\mathcal{T}_l$, respectively. Following Shinzato et al. (2022), the values in $\boldsymbol{v}_a^m$ or $\boldsymbol{v}_a^l$ are sorted in descending order according to their frequency.

Our proposed KSelF can efficiently reduce the size of the run-time knowledge exposed during the instance training, and can be applied with or without the pretraining corpus. During the training stage, KSelF inherits the advantage of Shinzato et al. (2022) by constructing all the four queries $\boldsymbol{q}_s$, $\boldsymbol{q}_u$, $\boldsymbol{q}_s^m$ and $\boldsymbol{q}_s^l$, which can exploit the run-time knowledge more efficiently to explain the meaning of the attribute, while reduce the dependence on the values in the run-time knowledge. During the inference stage, for a seen attribute, the run-time knowledge is a combination of the more-related knowledge and the less-related knowledge, and the query is built as:

$$\boldsymbol{q}_{s,test} = [\text{CLS}; \boldsymbol{x}; \text{SEP}; \text{SEEN}; \boldsymbol{a}; \text{SEP}; \boldsymbol{v}_a^m; \boldsymbol{v}_a^l];$$

while for an unseen attribute, the query is constructed as:

$$\boldsymbol{q}_{u,test} = [\text{CLS}; \boldsymbol{x}; \text{SEP}; \text{UNSEEN}; \boldsymbol{a}].$$

### 4.3 Pretraining with KSelF

In the pretraining stage, we first initialize KSelF with the a public pretrained model, e.g., BERT (Devlin et al., 2019), and then continue pretrain for the QA-based AVE task using the generated pretraining corpus introduced in Section 4.1. During the finetuning stage, the model is further tuned with any available downstream datasets.

## 5 Experiments

### 5.1 Experimental Setup

#### 5.1.1 Datasets

A widely adopted benchmark for the task of AVE is AE-pub (Xu et al., 2019). However, this dataset contains many noisy annotations and some dull attributes (e.g. "feature 1", "type"). These problems bring extra challenges for fairly evaluating an AVE system. To mitigate this problem, we propose a new dataset EC-AVE collected from a well-structured

| | AE-pub | | | EC-AVE | | |
|---|---|---|---|---|---|---|
| split | Train | Dev | Test | Train | Dev | Test |
| Total # | 76823 | 10975 | 21950 | 105625 | 15817 | 31876 |
| Pos. # | 59954 | 8518 | 17154 | 72719 | 10772 | 21897 |
| Neg. # | 16869 | 2457 | 4796 | 32906 | 5045 | 9979 |
| Attr. # | 1800 | 635 | 872 | 347 | 394 | 422 |
| Unseen. # | - | 154 | 246 | - | 58 | 75 |

Table 2: Statistics of the two datasets adopted in this work. "Pos.", "Neg.", "Attr.", and "Unseen." denote positive instance, negative instance, attribute, and unseen attribute, respectively.

E-commerce website. During the collection process, we guarantee that the EC-AVE dataset has no overlaps with the pretraining corpus to avoid data leakage. Table 2 shows statistics of two datasets.

#### 5.1.2 Implementation Details

Following Shinzato et al. (2022), we build our model upon a pretrained BERT in the uncased base version for fair comparisons. We first pretrain the model with the pretraining corpus for 5 epochs, with the maximum profile length as 64 and the maximum query length as 192. During the finetuning stage, we continually train from the pretrained model for 20 epochs on downstream datasets with the maximum profile length as 32 and maximum query length as 192. We follow the settings in Shinzato et al. (2022) for the rest hyper-parameters.

#### 5.1.3 Evaluation Metrics

We adopt precision, recall and F1 score as metrics. We follow Exact Match criteria (Rajpurkar et al., 2016) to compute the scores, where the full sequence of extracted value needs to be correct.

#### 5.1.4 Baseline Methods

**GPT-3.5** (Brown et al., 2020): We design a prompt to have GPT-3.5 to predict the values in a generative scheme. More details can be found in Appendix B.
**Dictionary**: We follow Shinzato et al. (2022) to perform a simple dictionary matching by returning the most frequent seen value included in the given title for a given attribute.
**SUOpenTag** (Xu et al., 2019): This is the method that achieves best performance under the sequence-tagging paradigm.
**BERT-QA** (Shinzato et al., 2022): This is the method that achieves the state-of-the-art performance on the AVE task[1].

---

[1] The code of BERT-QA is not publicly available, hence we report the performance of our reproduction.

| | AE-pub | | | | | | EC-AVE | | | | | |
|---|---|---|---|---|---|---|---|---|---|---|---|---|
| | Macro | | | Micro | | | Macro | | | Micro | | |
| | P | R | F1 | P | R | F1 | P | R | F1 | P | R | F1 |
| GPT-3.5 | 18.89 | 13.14 | 14.37 | 78.85 | 27.68 | 40.97 | 50.28 | 26.47 | 31.51 | 81.4 | 36.3 | 50.21 |
| Dictionary | 30.24 | 29.50 | 28.16 | 82.19 | 70.86 | 76.10 | 48.09 | 62.30 | 50.20 | 73.02 | 58.50 | 64.96 |
| SUOpenTag | 39.91 | 35.53 | 36.76 | 95.54 | 83.04 | 88.85 | 76.08 | 63.82 | 67.91 | 96.60 | 82.38 | 88.93 |
| BERT-QA | 46.24 | 42.47 | 43.23 | 93.66 | 85.54 | 89.42 | 76.41 | 69.14 | 71.05 | 94.89 | 83.99 | 89.11 |
| BERT-QA + pretrain | 47.02 | 42.81 | 43.69 | 93.69 | 85.85 | 89.60 | 78.84 | 70.33 | 72.99 | 95.67 | 85.82 | 90.48 |
| KSelF | 47.61 | 44.44 | 45.02 | 93.64 | 86.13 | 89.73 | 76.91 | 69.15 | 71.42 | 95.75 | 84.91 | 90.00 |
| KSelF + pretrain | **48.30** | **45.49** | **45.93**[*] | **94.06** | **86.28** | **90.00**[*] | **80.90** | **71.20** | **74.30**[*] | **96.32** | **86.15** | **90.95**[*] |

Table 3: Performance of different methods on AE-pub dataset and EC-AVE dataset. * means the gains are statistically significant with $p < 0.05$ compared with KSelF.

**BERT-QA+pretrain**: We first pretrain the BERT-QA model with our constructed pretraining corpus, and finetune on downstream datasets. The pretraining scheme and finetuning scheme follow the settings in Shinzato et al. (2022).

## 5.2 Experimental Results

### 5.2.1 Overall Performance

The results are presented in Table 3, and we report both macro and micro performance. There are mainly four observations we can conclude from the table. (1) Surprisingly, GPT-3.5 can achieve Macro/Micro F1 of 14.37/40.97 and 31.51/50.21 on AE-pub and EC-AVE respectively. It is capable of predicting the values in an unsupervised generative way to some extent, which demonstrates the potentials of large pretrained language models in solving the AVE task. (2) The QA-based methods significantly outperform the sequence-tagging-based SUOpenTag, showing the advantages of QA paradigm. (3) When the pretraining corpus is not exploited, our proposed KSelF outperforms BERT-QA and achieves new state-of-the-art performance on both two datasets. Such results demonstrate that the knowledge selection method can encourage the model to utilize the training data more efficiently, and can enhance the performance of the QA-based models. (4) The pretraining method can further boost the performance of KSelF. As shown in Table 3, when KSelF is firstly pretrained and then finetuned on downstream datasets, both Macro F1 and Micro F1 are improved while the improvement on Macro F1 is much more significant because of the long-tail distribution of the attributes. Such results consolidate the conclusion that the proposed pretraining method brings benefits to the AVE.

To more comprehensively evaluate the effectiveness of our constructed pretraining corpus and our proposed KSelF, we also explore to pretrain BERT-QA on the corpus. According to the performance reported in the Table 3, the pretraining boosts the performance of BERT-QA on both datasets. However, the improvement of BERT-QA is more marginal than that of KSelF, which further supports our motivation that the proposed knowledge-selective query expansion framework is capable of more efficiently exploiting the training data.

### 5.2.2 Results on Rare Attributes

We summarize the performance of BERT-QA and KSelF on attributes with different frequencies in Table 4. As reported in the table, on both AE-pub and EC-AVE, KSelF outperforms BERT-QA on frequent attributes (frequency $\geq 10$), while the performance is slightly worse than BERT-QA on unseen attributes. The reason is that KSelF can more efficiently utilize the values in the training data as run-time knowledge, while unseen attributes do not appear in the training data and thus the proposed KSelF has little effect on them since no run-time knowledge can be retrieved for unseen attributes.

The goal of pretraining is to enhance the model's generalizability and improve its performance on rare attributes. The results in Table 4 demonstrate that the performance of the two QA-based methods is significantly enhanced on unseen attributes. For the AE-pub dataset, Macro F1 and Micro F1 are lower for both methods on attributes with a frequency between 1 and 10 when pretraining is incorporated. The main reason is that AE-pub contains many grammatically wrong attributes and dull attributes, as shown in the examples in Appendix C. The pretraining corpus is not able to improve the generalizability on these attributes since the corpus are collected from E-commerce websites and the instances are well organized both literally and semantically. Meanwhile, we can see from the table that the improvement introduced by pretraining

| | AE-pub | | | | | | EC-AVE | | | | | |
|---|---|---|---|---|---|---|---|---|---|---|---|---|
| Frequency | High | | Low | | Unseen | | High | | Low | | Unseen | |
| Metrics | Ma-F1 | Mi-F1 | Ma-F1 | Mi-F1 | Ma-F1 | Mi-F1 | Ma-F1 | Mi-F1 | Ma-F1 | Mi-F1 | Ma-F1 | Mi-F1 |
| BERT-QA | 64.16 | 90.44 | 40.22 | 64.62 | 23.60 | 38.64 | 86.45 | 90.14 | 15.00 | 56.00 | 28.39 | 53.62 |
| BERT-QA + pretrain | 62.99 | 90.61 | 39.15 | 63.07 | 28.51 | 44.95 | 87.33 | 91.41 | 18.33 | 61.54 | 34.24 | 57.60 |
| KSelF | 64.89 | 90.73 | 44.48 | 67.28 | 22.57 | 38.10 | 87.25 | 91.05 | 13.33 | 52.17 | 27.75 | 53.25 |
| KSelF + pretrain | 65.15 | 90.99 | 42.52 | 65.95 | 28.95 | 45.81 | 87.89 | 91.82 | 21.67 | 64.29 | 37.90 | 60.47 |

Table 4: Performance on attributes with different frequencies. Ma-F1 and Mi-F1 denote Macro F1 and Micro-F1 respectively. *High* frequency means the number of occurrence of the attribute in the training data is greater than or equal to 10, while *low* frequency means the number is less than 10 but greater than 0. *Unseen* means the attribute never appear in the training data.

| | AE-pub | | EC-AVE | |
|---|---|---|---|---|
| | Ma-F1 | Mi-F1 | Ma-F1 | Mi-F1 |
| w/o KS | 35.92 | 66.17 | 48.62 | 64.33 |
| w/ KS | 38.36 | 69.01 | 49.47 | 66.44 |

Table 5: Zero-shot results on the two datasets. KS and Ma-F1/Mi-F1 denote knowledge selection and Macro/Micro F1 respectively.

on unseen attributes of KSelF is more substantial than that of BERT-QA, which further proves that when large-scale dataset is available, KSelF can utilize the pretraining data more efficiently and hence strengthen the model performance.

### 5.2.3 Zero-shot Results

We further evaluate the performance of our pretrained model under an extreme zero-shot setting. Specifically, we initialize the model with the pretrained checkpoints either with or without knowledge selection, and evaluate the model on both AE-pub and EC-AVE without any further finetuning. The results are reported in Table 5. As we can see, even if the model is not finetuned on the corresponding downstream datasets, it still exhibits the ability of generalization to some extent. Moreover, when pretraining step is performed without our proposed knowledge selection scheme, the zero-shot performance is much worse than pretraining with the proposed scheme. Such results also demonstrate the advantages of our proposed knowledge-selective query expansion framework.

### 5.3 Case Study

The primary benefit of pretraining is the improved generalizability on attributes with limited training resources. To have a better understanding about the proposed method, we present some examples in Table 6. The first example demonstrates that the pretraining can enhance the model's ability on predicting the rare values, e.g. "378037-623". Further investigations show that with pretraining, KSelF

**Profile**: Original official Nike Air Jordan 11 retro win like 96 men's basketball shoes sneakers sports AJ11 classic outdoor 378037-623

**Attribute**: model number  **Value**: 378037-623

**KSelF**: 378037  **KSelF+Pre.**: 378037-623

**Profile**: 1pcs crankbait fishing wobbler 14g 10cm artificial crank bait bass trout fishing lure pike trolling pesca minnow fishing tackle

**Attribute**: lure length  **Value**: 10cm

**KSelF**: 14g  **KSelF+Pre.**: 10cm

**Profile**: running street creative funny belly pockets outdoor sports zipper mobile phone pockets simulation butt anti-harassment waist bag

**Attribute**: opening method  **Value**: zipper

**KSelF**: belly pockets  **KSelF+Pre.**: zipper

Table 6: Example predictions of KSelF and KSelF with pretraining (Pre.) from AE-pub.

can improve its accuracy from 53.1% to 53.7% on the attribute "model number", of which the values are mostly rare tokens. Another advantage of pretraining is the enhancement of performance on unseen attributes. Here, "lure length" and "opening method" in the second and third examples are all unseen attributes as they do not appear in the training data. As we shown in the table, KSelF fails to predict their values while the pretraining method enables the model to correctly extractvalues from the product profiles.

## 6 Conclusions

Our study aims to improve the performance of Attribute Value Extraction for rare attributes, which is of significant importance for real-life applications. We propose a pretraining method to improve the generalization ability of the QA-based AVE model. In addition, we propose a knowledge-selective query expansion framework that can effectively exploit the training data. Our experimental results on both the AE-pub and EC-AVE datasets show that the proposed pretraining-based KSelF

achieves new SOTA performance.

## Limitations

Though our proposed KSelF has achieved the state-of-the-art performance and the performance is further improved with the pretraining corpus, the method can only tackle attributes/values appearing in the textual part of the product profile. It is often the case that some attributes and values appear in the non-text part, e.g. product image, while our proposed method does not cover such pairs. Although multimodal AVE also plays an important role in E-commerce, we do not discuss it this work and leave it for future work.

## Ethics Statement

Our proposed methods do not introduce any social/ethical bias to the model, while there are potentially concerns in the AE-pub dataset. We notice that some products are unisex according to the profiles while the annotation for the attribute "gender" is either "man" or "woman". Such problems in annotations may introduce gender bias in the training data, which will further influence the behaviors of the model. To avoid such problems, cleaner and better-formatted datasets are desired, and hence we build the EC-AVE dataset, which can provide fairer and more comprehensive evaluations to the AVE task.

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

## A Details of Similarity Calculation

We adopt different measurements of similarity for pretraining stage and fine-tuning stage for two reasons:

1. Entity retrieval for the E-Commerce domain is very time-consuming and model computation cost is very high. Considering the size of the pretraining corpus, we only apply entity retrieval during the fine-tuning stage.

2. We conduct comprehensive analyses over the bag-of-word (BoW) similarity. It turns out that we can achieve acceptable performance on the similarity calculation between different product profiles by applying BoW similarity. Considering the large size of the pretraining corpus, BoW similarity is a fairly good alternative to reduce the time and computation costs during the pretraining stage.

Specifically, for the entity similarity, we regard two profiles $x_i$ and $x_j$ and as similar profile if (1) $x_i$ and $x_j$ contain at least one common entity for the entity measurement, or (2) more than 1/3 of the tokens in $x_i$ appear in $x_j$ for the BoW measurement.

## B GPT-3.5 Implementation Details

We adopt GPT-3.5, specifically the model "gpt-3.5-turbo", to generate the values based on the given a product profile $x$ and an attribute $a$. We design different prompts and choose the one that achieves the best performance on the Dev set. The prompt we adopt is:

> Q: Given the product profile $x$, does the profile contain the value of the attribute $a$? If so, return the answer in the format of *(attribute, value)*; if not, return "No".
> A: ____

We have GPT-3.5 to generate the content in the blank, and the generated content will be parsed and the retrieved results are regarded as the corresponding predicted value.

## C Examples of Datasets

In this section, we will present some examples of the two datasets we use in this work. The current public benchmark AE-pub contains many instances of which the attributes are either grammatically wrong, like Example 2 and 3, or dull, like Example 4:

> Example 1
> **Profile**: adidas men shoes originals forum lo refined low-top men's skateboarding shoes cotton fabric adidas sports sneakers for men
> **Attribute**: brand name
> **Value**: adidas
>
> Example 2:
> **Profile**: handing overlength 8m-12m 13m 14m 15m high carbon super hard fishing rod telescopic rod sea fishing rod taiwan fishing rod
> **Attribute**: maerial (material)
> **Value**: carbon
>
> Example 3:
> **Profile**: outdoor mens hiking ski jacket camouflage thick warm assault jacket windproof camping jacket men plus size
> **Attribute**: patern (pattern)
> **Value**: camouflage
>
> Example 4:
> **Profile**: catch u 5-22g m/mh spinning casting rod carbon pole 1.8m 2 tips travel sea spinning fishing rod
> **Attribute**: feature 2
> **Value**: casting rod

In contrast, our proposed EC-AVE dataset is much better formatted and of larger size, and we present two examples below:

> Example 1
> **Profile**: huffy frozen 2 olaf preschool scooter , handlebar bin , three wheels & wide deck bullet point - brake style : foot bullet point - suspension type : rigid
> **Attribute**: brake style
> **Value**: foot
>
> Example 2:
> **Profile**: uxpro psd010bf heating only with fan digital thermostat bullet point - digital accuracy bullet point - battery powered only bullet point - large led illuminated digital display
> **Attribute**: model number
> **Value**: psd010bf

## D Examples of E-commerce

In Figure 4, we present three website examples that contain product descriptions and the corresponding specifications.

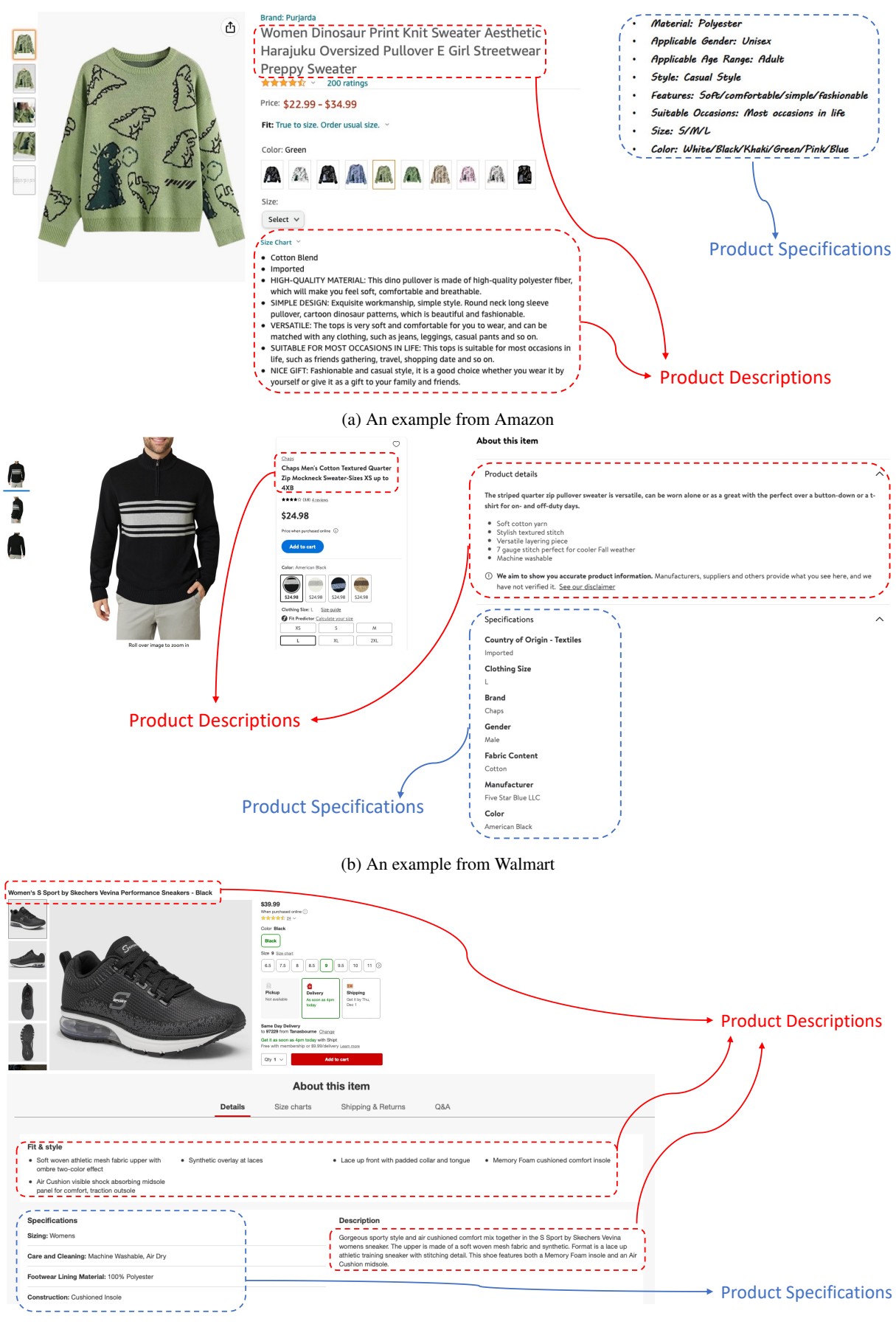

Figure 4: Examples from various E-commerce websites that contain product descriptions and the corresponding specifications.