# OpenReview forum: "Knowledge-Selective Pretraining for Attribute Value Extraction"
_EMNLP/2023/Conference — EMNLP 2023 Findings_

### Official Review · Reviewer_Z4E8 · 2023-07-30

**Soundness:** 4

**Excitement:**

4: Strong: This paper deepens the understanding of some phenomenon or lowers the barriers to an existing research direction.

**Paper Topic And Main Contributions:**

This paper proposed a knowledge selection (KS) framework and pre-training strategy for attribute-value extraction task. They use the KS framework to expand the query more accurately. They created a pre-training corpus for this task from E-commerce sites. They pre-train the BERT on this corpus along with KS framework and during fine-tuning on task specific dataset also, they use this KS framework. Experiments on the noise AE-pub and their own EC-AVE datasets show the effectiveness of the proposed work.

**Reasons To Accept:**

1. Authors proposed an pre-training corpus creation strategy for AVE task and showed the effectiveness of such corpus and pre-training on BERT and KSelF models.
2. Although overall performance improvement is not very high on AE-pub or EC-AVE datasets, improvements on the low-frequency attributes and unseen attributes are quite significant on these two datasets.


**Reasons To Reject:**

1. The experiments section of the paper may be considered little weak as the AE-pub dataset is noisy as mentioned by the authors. The experiment section would have been more robust if they included MAVE ([1]) dataset as well.


[1] MAVE: A Product Dataset for Multi-source Attribute Value Extraction

**Reproducibility:**

4: Could mostly reproduce the results, but there may be some variation because of sample variance or minor variations in their interpretation of the protocol or method.

**Reviewer Confidence:**

4: Quite sure. I tried to check the important points carefully. It's unlikely, though conceivable, that I missed something that should affect my ratings.

---

> ### Author Rebuttal · Authors · 2023-08-29
>
> Thanks for your comments!
>
> ## Comment 1
>
> > Comment 1.1: "The experiment section would have been more robust if they included MAVE ([1]) dataset as well."
>
> Firstly, we construct and release a new benchmark dataset EC-AVE which is larger and cleaner than the AE-pub dataset. Experiments on both the datasets show that our method is effective.
>
> Secondly, the goal of the MAVE dataset is different from ours. MAVE aims to retrieve attributes from multiple sources like product descriptions, which is very different from our setting where the product profile is the product title.  AVE on product titles is not easy as the product titles are more concise and informative, and sometimes the titles are not coherent natural language. Harvesting attributes/values from product titles is a more challenging task compared to the setting in MAVE dataset. Meanwhile, as reported in the MAVE paper, the performance of the BERT-QA baseline on the MAVE dataset is 98.14, which is very saturated (some researchers have also reported that the MAVE dataset is noisy and too simple), while the Micro-F1 scores achieved by BERT-QA on the two datasets adopted in this paper are 89.42 and 89.11 respectively. The lower numbers further show that AVE on product titles is more challenging than the setting in the MAVE dataset.

---

### Official Review · Reviewer_a4W9 · 2023-08-04

**Soundness:** 4

**Excitement:**

3: Ambivalent: It has merits (e.g., it reports state-of-the-art results, the idea is nice), but there are key weaknesses (e.g., it describes incremental work), and it can significantly benefit from another round of revision. However, I won't object to accepting it if my co-reviewers champion it.

**Missing References:**

Li Yang, Qifan Wang, Zac Yu, Anand Kulkarni, Sumit Sanghai, Bin Shu, Jon Elsas, and Bhargav Kanagal. 2022. MAVE: A Product Dataset for Multi-source Attribute Value Extraction. In Proceedings of the Fifteenth ACM International Conference on Web Search and Data Mining (WSDM '22). Association for Computing Machinery, New York, NY, USA, 1256–1265. https://doi.org/10.1145/3488560.3498377

**Paper Topic And Main Contributions:**

This paper presents methods to improve QA-based models for the Attribute Value Extraction (AVE) task, especially for rare attributes. The main contributions of this paper include:
1. a Knowledge-Selective Framework (KSelF) that can use training data more efficiently, and a pre-training method for QA-based AVE models to improve model performance on rare and unseen attributes,
2. EC-AVE, a new well-structured benchmark for the AVE task, and
3. quantitative experimental results showing that the proposed methods outperform SOTA.

**Questions For The Authors:**

Question A: This paper pointed out that attribute values for the same attribute on products from different categories are in relatively disjoint space as a motivation for selecting knowledge when performing query expansion (i.e., categorizing values into more and less related). However, since product profiles are often associated with categories (or can be classified if not), the same can be done by retrieving values for query expansion within products of the same category. Are there any other properties of product profiles that KSelF could disambiguate to improve performance?

**Reasons To Accept:**

1. The newly proposed method, based on QA-AVE approach, is well grounded in current literature and addresses drawbacks of previous work.
2. The experiment comparing the proposed method with SOTA models validates the performance gain of KSelF and the value of the pre-training corpus.

**Reasons To Reject:**

1. The pre-training method relies on psudo-labeling based on a naive approach (extract string match) and is not a novel contribution per se.
2. The benchmark contribution is compared to a relatively old and smaller dataset, while a newer, larger, and cleaner open dataset is available.

**Reproducibility:**

5: Could easily reproduce the results.

**Reviewer Confidence:**

5: Positive that my evaluation is correct. I read the paper very carefully and I am very familiar with related work.

---

> ### Author Rebuttal · Authors · 2023-08-29
>
> Thanks for your comments!
>
> ## Comment 1
>
> > Comment 1.1: "The pre-training method relies on psudo-labeling based on a naive approach (extract string match) "
>
> Our main contributions in this work include: (1) We are the first to propose pretraining on the AVE task. (2) We design a Knowledge-Selective Framework that can be closely combined with pretraining to effectively improve the overall performance, as well as the performance on the rare and unseen attributes. (3) We construct and release a new benchmark dataset that can encourage future research in this field.
>
> Constructing a task-specific pretraining corpus is expensive and time-consuming, especially when the pretraining corpus is particularly large. Existing works mostly adopt pseudo-labeling methods to save the cost on the construction. Although our adopted approach is simple and not convoluted, it is shown to be very effective according to the experiments, and we consider that simple and effective models are often preferred in real-life applications.
>
> ## Comment 2
>
> > Comment 2.1: The benchmark contributions as a newer, larger, and cleaner open dataset is available.
>
> Firstly, we construct and release a new benchmark dataset EC-AVE which is larger and cleaner than the AE-pub dataset. Experiments on both two dataset show that our method is effective.
>
> Secondly, we are not sure which newer dataset is referred to. If the MAVE dataset is referred to, the goal of this dataset is different from ours. MAVE aims to retrieve attributes from multiple sources like product descriptions, which is very different from our setting where the product profile is the product title. AVE on product titles is not easy as the product titles are more concise and informative, and sometimes the titles are not coherent natural language. Harvesting attributes/values from product titles is a more challenging task compared to the setting in the MAVE dataset. Meanwhile, as reported in the MAVE paper, the performance of the BERT-QA baseline on the MAVE dataset is 98.14, which is very saturated (some researchers have also reported that the MAVE dataset is noisy and too simple), while the Micro-F1 scores achieved by BERT-QA on the two datasets adopted in this paper are 89.42 and 89.11 respectively. The lower numbers further show that AVE on product titles is more challenging than the setting in the MAVE dataset.
>
> ## Question
>
> > Question A: "Are there any other properties of product profiles that KSelF could disambiguate to improve performance?"
>
> Yes. Our proposed method can be extended if there are more intermediate categories. In our work, we classify the run-time knowledge as the more-related and the less-related. However, if we have more fine-grained supervision or supervision detection tools, e.g., the product type the current profile is about, we can design more categories for the run-time knowledge and extend our method accordingly.  Thank you for this constructive question, and we will explore this in our future study.

---

### Official Review · Reviewer_YCV3 · 2023-08-05

**Typos Grammar Style And Presentation Improvements:** Line 321
**Soundness:** 4

**Excitement:**

2: Mediocre: This paper makes marginal contributions (vs non-contemporaneous work), so I would rather not see it in the conference.

**Missing References:**

MAVE: A Product Dataset for Multi-source Attribute Value Extraction

**Paper Topic And Main Contributions:**

The paper proposes a new method for attribute value extraction (AVE) called Knowledge-Selective Pretraining for Attribute Value Extraction (KSelF). KSelF addresses the limitations of existing AVE methods by leveraging pretraining and query expansion. The pretraining method is based on a large-scale corpus of product profiles and attributes collected from E-commerce websites. The query expansion method uses a knowledge-selective framework to select informative knowledge from the pretraining corpus to expand the query. The experiments on two benchmark datasets show that KSelF achieves state-of-the-art performance on both datasets. KSelF also outperforms existing methods on rare and unseen attributes.

**Questions For The Authors:**

* This work basically applies distant supervision to construct pretraining dataset. Is the precision of distant supervision100%? Does tge wrong product problem occur, i.e. the profile extracted is not related to the product of interest?

**Reasons To Accept:**

* KSelF tries to address the limitations of existing methods. KSelF first categorizes the knowledge (i.e., attribute values) into more-related and less-related ones, which further uses the seen values of an attribute to expand the query.
* KSelF achieves state-of-the-art performance on two benchmark datasets.
* KSelF outperforms existing methods on rare and unseen attributes.

**Reasons To Reject:**

* The work is purely augmented from Shinzato et al. (2022) without significant innovation. The improvements from the baselines are very limited, e.g. from 89.60 micro-F1 to 90.00. GPT-3.5's performance highly relies on how the prompt was designed. It may not be fair to report its performance according to one version of prompt.

* Calculating similarity scores between profiles seems to be the key to KSelF. It's not clear that how entity retrieval works for calculating the similarity in the fine-tuning stage, e.g. what are the entities in the profile? can we retrieve attributes instead? how time-consuming is it? what's the performance without this entity retrieval? And the similarity measurement in pretraining stage is based on bag-of-word similarity, which may not be accurate enough to measure the similarity between two profiles. This could lead to the selection of irrelevant knowledge to expand the query.

* As mentioned by the author, the proposed method is designed to solve text-only AVE task. Also, the method can only deal with cases where attribute values can be explicitly found in the text. For example, the attribute is water_resistant and value is True, which should be able to be inferred from the profile: "water resistant up to 50 meters". Overall the impact and coverage of this work is limited.

**Reproducibility:**

3: Could reproduce the results with some difficulty. The settings of parameters are underspecified or subjectively determined; the training/evaluation data are not widely available.

**Reviewer Confidence:**

4: Quite sure. I tried to check the important points carefully. It's unlikely, though conceivable, that I missed something that should affect my ratings.

---

> ### Author Rebuttal · Authors · 2023-08-29
>
> Thanks for your comments.
>
> ## Comment 1
>
> > Comment 1.1: contributions w.r.t, previous work (Shinzato et al., 2022)
>
> We consider that innovation and novelty connote different respects. Our work is based on intuitive motivation and we are the first to propose the pretraining method for attribute value extraction.  While the proposed method is not convoluted, it outperforms all prior methods and achieves SOTA performance on the benchmark datasets. Simple and effective models are often preferred in real-life applications. In addition, our method is shown to be effective also on the rare and unseen attributes, which has not been studied in previous works but is important for many applications.
>
> Second, the Knowledge Selective Framework can be closely combined with pretraining. As shown in Table 3 and Table 4, the baseline method proposed by Shinzato et al. (2022) cannot be effectively applied in pretraining due to the lack of fine-grained run-time knowledge categorization, while KSelF is shown to be effective with or without pretraining according to Table 3.
>
> Thirdly, we collect and release a new benchmark dataset which is larger and better-formatted compared with the previous benchmark. This benchmark can help with the future research in this field. We take it as another major contribution of our work.
>
> > Comment 1.2: Improvements over the baselines
>
> We note that the proposed KSelF achieves improvement over all the metrics on both of the datasets. The improvement is statistically significant with p < 0.05. We will add the significance test results in revision. Thank you for your suggestions.
>
> According to Table 3, KSelf+pretrain improves the previous SOTA performance from 43.23 to 45.93 on Macro-F1 and from 89.42 to 90.00 on Micro-F1 on the AE-pub dataset, and from 71.05 to 74.30 on Macro-F1 and from 89.11 to 90.95 on Micro-F1 on the EC-AVE dataset. Such improvement is substantial. Even if we take BERTQA+pretrain as a comparison, our method outperforms it by 43.9 to 45.93 on Macro-F1 and from 89.60 to 90.00 on Micro-F1 on the AE-pub, and 72.99 to 74.30 on Macro-F1 and from 90.48 to 90.95 on Micro-F1 on the EC-AVE dataset.
>
> Note that among different metrics, the reason why the improvement is less significant on the Micro-F1 is that Micro-F1 takes all the true-positives into calculation. Both datasets exhibit a long-tail distribution on attributes, so there are more examples on the frequently seen attributes, which makes the absolute improvement on Micro-F1 less significant. The similar phenomenon can be observed in previous works such as Shinzato et al. (2022) as well.
>
> > Comment 1.3: Fairness to report GPT 3.5's performance according to one version of prompt.
>
> As stated in Appendix B, we design different prompts and choose the one that achieves the best performance on the development set for GPT-3.5. Hence the performance of GPT-3.5 is well validated. We assume that there might be better methods that can achieve good performance with GPT-3.5, but it is out of the scope of this paper. We leave such a problem as future work on how to build AVE frameworks based on LLMs.
>
> ## Comment 2
>
> > Comment 2.1: How entity retrieval works for calculating the similarity in fine-tuning.
>
> The entity retrieval aims to categorize the run-time knowledge into more-related or less-related. For this step, we utilize two different ways, as shown in Appendix A. In brief, we adopt the entity similarity during finetuning, while adopting bag-of-word similarity during pretraining.
>
> > Comment 2.2: "what are entities" and "can we retrieve attributes instead"
>
> Our setting is to extract attribute values from product profiles (specifically product titles). We adopt the entity retrieval tool (De Cao et al., 2021). So the entities retrieved from the profile mostly include brands (Nike, Samsung, etc.), item names (shoe, cloth, etc.), human beings (boy, woman, etc.). These entities well present the main topic of a product profile, and hence can be used for calculating the similarities between product profiles.
>
> As you can see, we cannot retrieve attributes instead. The attributes cannot represent the topic of a product profile because different product profiles may contain the same attributes. For example, a product related to "cloth" and a product related to "electronic" may both contain the attribute "brand", but they cannot serve as effective run-time knowledge for each other.
>
> > Comment 2.3: "how time-consuming is entity retrieval"
>
> The goal of entity retrieval is to detect the similarity between two product profiles, so we want the entity retrieval tool to be universal and open-domain. We adopt the tool from De Cao et al., 2021 in our paper. For this tool, it took around 3 seconds to finish a single example, which is pretty time-consuming.
>
> > Comment 2.4: "what's the performance without this entity retrieval?"
>
> Our method is built upon the categorization of more-related knowledge and less-related knowledge, while entity retrieval is adopted to satisfy this goal. If no entity retrieval is adopted, then our method no longer stands. Or if we remove the categorization step, KSelF will degrade to BERT-QA. Thank you so much for the comments, we will make this clearer.
>
> > Comment 2.5: "which may not be accurate enough to measure the similarity between two profiles."
>
> The product titles are very different from normal language. The product titles are often informative yet concise, and the tokens included are mostly related to certain aspects of the product. As we are conducting attribute value extraction on product titles, the bag-of-word similarity works pretty well if the threshold is reasonable and carefully selected. We do not have golden labels to evaluate the performance of this step, but human checks over 30 random samples during our experiments on each dataset show that averagely less than 2 out of 14 values are "less-related" to the current product profile, which indicates that the performance of bag-of-word similarity is high enough to satisfy our goal.
>
> Meanwhile, we want to point out that the goal of categorization for more-related knowledge and less-related knowledge is to encourage the model to exploit the run-time knowledge more effectively, and hence a 100% accuracy is not required.
>
> ## Comment 3
>
> > Comment 3.1: "the proposed method is designed to solve text-only AVE task."
>
> The text-based AVE is a well-established task and has been widely studied for years. In this paper, we strictly follow the setting employed by previous research to solve the text-based AVE task, i.e., retrieving attributes/values from text.
>
> > Comment 3.2: only deal with cases where attribute values can be explicitly found in the text
>
> We follow previous works by solving the AVE task from the retrieval-based scheme. We want to point out that existing works on the AVE task mostly focus on the retrieval-based scheme, by either adopting a sequence-tagging paradigm or a QA paradigm. Both of the two paradigms mainly deal with values that are explicitly presented in the text. For the attributes that are implicitly presented, a different paradigm is required which is out of the scope of this paper.
>
> > Comment 3.3: Overall the impact and coverage of this work.
>
> Our work strictly follows the problem setting that is well-established in previous works, which is also the setting widely adopted in current research, where AVE is conducted by retrieving attributes/values from text and the attributes/values are explicitly presented in the text. Considering the large number of published works on this task previously and the impact of solutions, we do not agree with the claim that the impact and coverage are limited.
>
> ## Question
>
> > Question 1: "This work basically applies distant supervision to construct pretraining dataset. Is the precision of distant supervision100%? Does the wrong product problem occur, i.e. the profile extracted is not related to the product of interest?"
>
> Firstly, we point out that the precision of distant supervision can never reach 100%. The goal is to categorize the run-time knowledge into the more-related and the less-related by our designed similarity measurements. So there will be errors occurring since the similarity measurements are unsupervised. But such errors are not frequent since we carefully design the entity-based similarity measurement and the bag-of-word-based similarity measurement. Experiments show that our method works and is effective.
>
> Meanwhile, even if all the intermediate categorization is wrong, our method will degrade to the baseline method of BERT-QA, which is the previous SOTA method.

---

### Meta-Review · Area_Chair_TVSa · 2023-09-20

**Recommendation:** 2

**Metareview:**

The paper proposes a new method for attribute value extraction that addresses the limitations of existing methods by leveraging pre-training and query expansion. The experiments on two benchmark datasets show it achieves state-of-the-art performance on both datasets and outperforms existing methods on rare and unseen attributes.

This paper makes some marginal but valuable contributions that are explained through a long rebut process. However, such contributions are still just incremental to existing literature and they still leave the paper lacking in novelty.

---

### Decision · Program_Chairs · 2023-10-07

**Decision:**

Accept-Findings

**Comment:**

The paper proposes a new method for attribute value extraction that addresses the limitations of existing methods by leveraging pre-training and query expansion. The experiments on two benchmark datasets show it achieves state-of-the-art performance on both datasets and outperforms existing methods on rare and unseen attributes.

This paper makes some marginal but valuable contributions that are explained through a long rebut process. However, such contributions are still just incremental to existing literature and they still leave the paper lacking in novelty.